# Maternal Stress and Excessive Weight Gain in Infancy

**DOI:** 10.3390/ijerph19095743

**Published:** 2022-05-09

**Authors:** Katelyn Fox, Maya Vadiveloo, Karen McCurdy, Sara E. Benjamin-Neelon, Truls Østbye, Alison Tovar

**Affiliations:** 1Department of Nutrition and Food Science, University of Rhode Island, Kingston, RI 02881, USA; maya_vadiveloo@uri.edu; 2Department of Human Development and Family Science, University of Rhode Island, Kingston, RI 02881, USA; kmccurdy@uri.edu; 3Department of Health, Behavior and Society, Johns Hopkins Bloomberg School of Public Health, Baltimore, MD 21205, USA; sara.neelon@jhu.edu; 4Department of Community and Family Medicine, Duke University Medical Center, Durham, NC 27710, USA; truls.ostbye@duke.edu; 5Department of Behavioral and Social Sciences, Brown University, Providence, RI 02912, USA; alison_tovar@brown.edu

**Keywords:** rapid weight gain, infancy, maternal mental health, perceived stress

## Abstract

Rapid weight gain in infancy increases the risk of developing obesity early in life and contributes significantly to racial and ethnic disparities in childhood obesity. While maternal perceived stress is associated with childhood obesity, little is known about the impact it has on infant weight gain. Therefore, this study explores the impact of maternal perceived stress on change in weight-for-length (WFL) z-scores and the risk of rapid weight gain in infancy. We conducted a secondary data analysis of the longitudinal Nurture birth cohort (*n* = 666). Most mothers in the cohort were non-Hispanic/Latinx Black (71.6%). About one-half of mothers had a body mass index (BMI) greater than 25 prior to pregnancy, were unemployed, and had a low income. Most infants in the cohort were born full-term and were of normal weight. Data were collected at 3-, 6-, 9-, and 12-months postpartum. At each assessment, mothers completed the Cohen’s Perceived Stress Scale (PSS), and research assistants weighed and measured each infant. Tertiles were used to compare mothers with high and low perceived stress. A mixed model analysis of repeated measures assessed the associations between baseline perceived stress and the change in infant WFL z-scores over time. Log-binomial models assessed the association between baseline perceived stress and rapid weight gain, defined as a change in WFL z-score > 0.67 standard deviations from three to twelve months. Just under one-half of the infants (47%) experienced rapid weight gain between three and twelve months of age. Birthweight for gestational age (RR = 1.18, 95% CI = 1.08–1.29, *p*-value = 0.004), gestational age at birth (RR = 1.07, 95% CI = 1.01–1.14, *p*-value = 0.031), and weeks breastfed (0.99, 95% CI 0.99–1.00, *p*-value 0.044) were associated with risk of rapid weight gain in unadjusted analyses. WFL z-scores increased significantly over time, with no effect of perceived stress on change in WFL z-score or risk of rapid weight gain. Rapid weight gain in infancy was prevalent in this sample of predominately Black infants in the Southeastern US. We did not find evidence to support the hypothesis that maternal perceived stress influenced the risk of rapid weight gain. More work is needed to identify and assess the risk factors for rapid weight gain in infancy and to understand the role that maternal stress plays in the risk of childhood obesity so that prevention efforts can be targeted.

## 1. Introduction

Childhood obesity continues to be a significant public health issue in the US, with clear disparities in prevalence and severity across race, ethnicity, and socio-economic status that begin early in life [1,2]. Addressing obesity early in life is critical as it tracks into adulthood and is associated with significant comorbidities, including cardiometabolic disease [3,4]. One factor that is shown to contribute to the overall risk of developing childhood obesity, as well as to differences among racial and ethnic groups, is rapid weight gain in infancy.

Rapid weight gain, commonly defined as a change in the weight-for-age (WFA) or weight-for-length (WFL) z-scores of >0.67 standard deviations, is demonstrated by a cross in percentile bands on the World Health Organization growth chart and has been associated with increased odds of obesity in childhood, adolescence, and young adulthood worldwide [5,6,7]. This association between rapid weight gain and later obesity risk remains independent of birth weight, except for preterm (<34-week gestational age] infants [8]. Importantly, data from the Early Childhood Longitudinal Study (ECLS)—Birth Cohort showed that weight gain in infancy was the largest contributing factor to differences in childhood obesity between racial and ethnic groups [9]. The rate of weight gain in infancy accounted for 40–70% of the disparities in childhood obesity at kindergarten entry between White and Black children. Understanding contributors to rapid weight in infancy may be important for addressing disparities in childhood obesity. 

Studies have identified both maternal and infant factors that impact rapid weight gain in infancy, including non-modifiable (gender, race, ethnicity, and parity) as well as some modifiable factors (gestational weight gain, gestational age, low birth weight, formula feeding, feeding on a schedule, maternal cigarette smoking) [10,11,12,13]. Although several risk factors have been identified, they account for a small percentage of rapid weight gain variance (7–11%), suggesting that other factors, such as maternal stress, may be playing an important role [10]. Maternal stress early in her child’s life has been shown to increase obesity risk from toddlerhood to adolescence [14,15]. There are several proposed mechanisms by which perceived stress in mothers impacts child BMI including having lower involvement with children, being less responsive to children’s needs, and modeling unhealthy behaviors [16,17,18,19]. High levels of perceived stress (perceiving one’s life as stressful, unpredictable, uncontrollable, and overloaded) may negatively impact maternal–child relations by decreasing how well a mother attunes to cues from her child [20] and the child’s secure attachment, which influences the child’s obesity risk [18]. For example, one study found that a poor maternal-child relationship in early childhood (15–36 months) was associated with 2.45 times higher odds of adolescent obesity [18]. Maternal perceived stress has also been associated with higher energy intake and increased consumption of breads and cereals among 0- to 6-month-old infants [21], although findings on the relationship between maternal stress and diet throughout childhood are less consistent [22]

Even though maternal perceived stress in the first year of life has been associated with childhood obesity risk and infant feeding, few studies have explored the impact of maternal perceived stress on infant weight gain trajectories and rapid weight gain to determine if stress impacts child obesity risk earlier than previously thought [13]. This question is particularly important to explore in populations exposed to chronic stressors such as racism or financial insecurity. Therefore, the goal of this study is to explore the association between perceived stress, change in WFL z-score over time, and rapid weight gain (increase in WFL z-score > 0.67) in infancy in a predominately low-income Black cohort. We hypothesized that higher maternal perceived stress would be associated with increases in WFL z-score over time and more rapid infant weight gain between three and twelve months. 

## 2. Materials and Methods

### 2.1. Study Design and Subject Characteristics

This study was a secondary data analysis of the longitudinal Nurture birth cohort (*n* = 666), which aimed to evaluate the impact of caregivers on infant adiposity and weight trajectories through the first year of life [23]. Mothers were 20–36 weeks’ gestation at the time of recruitment, pregnant with a singleton with no known congenital abnormalities, >18 years of age, able to speak and read English, intending to keep the baby and reside in the area (North Carolina, USA) at least 12 months postpartum. After birth, researchers excluded subjects if infants were born prior to 28 weeks of age, with congenital abnormalities that could affect growth and development, were hospitalized more than three weeks after birth, or were not feeding by mouth at the time of hospital discharge. Trained research staff collected data on maternal demographics and infant anthropometric measurements during home visits when infants were 3, 6, 9, and 12 months. Data collection took place from 2013 to 2017. The study was approved by the Institutional Review Board of Duke University Medical Center and full study details have been previously published [23].

### 2.2. Exposure Variable

Mothers completed the Perceived Stress Scale (PSS) [24], which is a 10-item scale that measures “the degree to which individuals appraise situations in their lives as stressful” on a five-point Likert scale ranging from never (0) to very often (4). Survey items are available in the Appendix A. A systematic review found the PSS to have good internal consistency (α = >0.7 in 12 studies), and high test–retest reliability (>0.7 in four studies) [25]. The measure has demonstrated criterion validity with other measures of stress and depression and predictive validity with groups known to have higher levels of perceived stress [25,26,27,28]. Four of the ten items were reverse-scored, and all of the items’ scores were summed to create a total score with possible scores ranging from 0–40, with a higher score indicating higher perceived stress. PSS was measured at baseline (three months postpartum) and at each follow-up visit (6, 9, and 12 months postpartum) and had strong internal validity in this sample with a Cronbach alpha of 0.89. PSS was assessed continuously, and tertiles were used to compare the risk of rapid weight gain in those with high(Q3) compared to low (Q1) stress, as has been conducted in previous studies [29]. 

### 2.3. Outcome Variable

Research staff, trained in standardized methods of anthropometric assessment, obtained measured infant’s weight and length in participants’ homes [23]. We then calculated z-scores using World Health Organization age and sex-specific references [30]. Rapid weight gain is a binary outcome, defined as an infant with an increase in weight-for-length z-score > 0.67 SD from three months postpartum to study completion (12 months postpartum) [31]. 

### 2.4. Covariates

We selected covariates measured during the Nurture demographic survey with established relationships with the exposure and outcome [10,32]. Covariates included mother’s age, race, pre-pregnancy BMI, prenatal diet quality, marital status, education, income, household composition and the infant’s birth weight, gestational age at birth, gender, race, ethnicity, and total weeks the infant was breastfed. Mothers completed the Block FFQ, which assesses diet over the last 30 days [33]. We assessed prenatal diet quality using the Alternate Healthy Eating Index 2010 (AHEI-2010), excluding the alcohol category, scored from 0 to 100, with higher scores indicating better diet quality [34].

### 2.5. Statistical Analysis

We conducted analyses using SAS version 9.04 for Linux (SAS Institute Inc., Cary, NC, USA). We assessed data for missingness. All the time-varying variables had missing data, as is common in longitudinal research studies (Appendix A) [35]. We flagged missing data and assessed differences between observations with and without missing exposure (PSS score) using independent t-tests for normally distributed continuous variables, Wilcoxon rank-sum test for non-normally distributed continuous variables, and Chi-Square tests for categorical variables (Appendix A). Women who had missing data were significantly younger and more likely to be Black, less likely to be Hispanic/Latinx, married or living with a partner, and have a high school education or greater. Infants with missing data were more likely to be Black, less likely to be Hispanic/Latinx, and were breastfed for fewer weeks. We determined, based on the limited number of significant associations with missingness, that the data were missing at random. Therefore, we used multiple imputations using the time-varying exposure and outcome variable, covariates, and auxiliary variables associated with missingness [35,36,37,38]. We used a fully conditional specification algorithm for imputation, with logistic regression used for binary variables (rapid weight gain, education, had overweight/obesity pre-pregnancy, low-income, unemployed, marital status, and household composition) and predictive mean matching used for non-normally distributed variables (PSS score, household composition, and AHEI-2010 score). Given the high proportion of missing data, we created 50 imputations [35]. All subsequent analyses were conducted using the imputed datasets and the estimates, and the confidence intervals presented reflect the pooled results [38].

First, the stability of PSS over timed was assessed using Pearson’s correlation between timepoint pairs and intraclass correlation coefficients calculated using a random intercept linear mixed model. PSS scores were moderately correlated across all timepoints (r 0.5–0.6, *p* < 0.001). The ICC was 0.6, indicating good stability across time *p* < 0.001. We chose to assess baseline PSS tertiles (three-months postpartum) in all regressions to establish temporality between exposure and outcome, given the stability of the PSS measure over time. We calculated means and standard errors for continuous variables and percentages for categorical variables for the sample and each PSS tertiles. We assessed for trends in baseline demographic characteristics across PSS tertiles using linear regression for continuous variables and Cochran–Mantel–Haenszel test for categorical variables [29]. 

Given the high prevalence of rapid weight gain, odds ratios may not be a good approximation of the risk ratio. Therefore, log-binomial models were developed to assess the association between risk factors (PSS and covariates) and rapid weight gain. We calculated the mean change in WFL z-score at each timepoint by baseline PSS tertiles. We then fit a mixed model analysis of repeated-measure using a compound symmetry covariance structure to assess the association of PSS tertiles and the change in WFL z-score accounting for between and within-subject variation. 

To assess the association between tertiles and rapid weight gain (change in WFL z-score > 0.67), we developed Log-binomial models. All covariates were added individually to the model assessing the influence of perceived stress tertiles on risk for rapid weight gain and kept in the adjusted model if the estimate changed by 10% or greater. Given that previous literature suggests that gender, race, and income status may be potential moderators between maternal perceived stress and childhood obesity risk, we tested for interaction effects between these variables and PSS.

## 3. Results

Demographic and anthropometric characteristics of the study population, along with the imputed values, are provided in Table 1. At baseline, women were on average 27.3 +/− 5.7 years, primarily non-Hispanic/Latinx (93.5%), Black (71.5%), with poor diet quality (AHEI-2010 42.2 ± 11.2). About one-half of women had overweight or obesity prior to pregnancy (61.4%), were married or living with a partner (57.5%), were unemployed (50.9%), and had a low income (<$20,000/year) (57.6%). On average, infants were full-term, born at a normal weight for gestational age (z-score −0.3 +/− 0.9), and were breastfed on average for 14.7 +/− 18.2 weeks. 

On average, at three-months postpartum, the PSS score was 12.8 ± 0.33, with a range of 0–30. When divided into tertiles, the low-stress group had a mean of 5.3 ± 0.2 and a range of 0–9, the moderate-stress group had a mean of 12.3 ± 0.1 and a range of 10–15, and the high-stress group had a mean of 20.7 ± 0.4 and a range of 16–39. There were no significant differences in demographic or anthropometric characteristics across PSS tertiles (Table 1). 

Just under one-half of the infants (47%) experienced rapid weight gain between baseline (three months) and follow-up (12 months). Certain covariates were associated with rapid weight gain, including birthweight-for-gestational-age z-score and gestational age at birth (Appendix A). A 1 standard deviation increase in birth weight-for-age z-score was associated with 1.18 times greater risk of rapid weight gain (RR = 1.18, 95% CI = 1.08–1.29, *p*-value < 0.001). A one-week increase in gestational age at birth was associated with 1.07 times greater risk of rapid weight gain (RR = 1.07, 95% CI = 1.01–1.14, *p*-value = 0.031). Total weeks breastfeeding was also associated with risk of rapid weight gain between 3 and 12 months (RR = 0.99, 95% CI = 0.99–1.00, *p*-value = 0.044), whereby breastfeeding for one year was associated with a 27 percent lower risk of rapid weight gain (RR = 0.73, 95% CI = 0.54–1.00, *p*-value = 0.049).

The weight-for-length z-score increased significantly over time with no significant effect on the baseline PSS tertiles. (Figure 1, Table 2). PSS and PSS tertiles were not significantly associated with an increased risk of rapid weight gain in the unadjusted or adjusted models (Table 3). No significant interaction effects were present for gender, race, or income status.

## 4. Discussion

The goal of this study was to assess the impact of perceived stress on infant weight gain trajectories and rapid weight gain among this predominately Black, low-income sample. We found that PSS scores were low and stable over time. Rapid weight gain in infancy was prevalent in this population, with nearly half of infants experiencing rapid weight gain from 3 to 12 months postpartum. Contrary to our hypothesis, we did not find any association between the mother’s perceived stress and the change in WFL z-score over time or infant rapid weight gain. More research is needed to identify and assess predictors for rapid weight gain in infancy. 

The primary results of this study suggest that there is insufficient evidence to claim that perceived stress in mothers influences an infants’ weight gain trajectories or risk of rapid weight gain in the first year of life. These results are consistent with recent studies of the Maternal Adiposity, Metabolism, and Stress (MAMAS) Study, a non-randomized control trial [13,39]. In this study of racially and ethnically diverse mothers and their infants, researchers found that neither prenatal nor postpartum perceived stress nor depressive symptoms were associated with rapid weight gain in infancy [13]. This study found, however, that the number of stressful life events a woman experienced in pregnancy was associated with 40% greater odds of rapid weight gain from birth to six months. In addition, they also found that mothers with higher perceived stress rated their infants as more reactive and less able to regulate but perceived stress was not associated with the infants’ physiologic response to stress. In contrast, mothers who experienced more stressful life events in pregnancy had infants with greater physiological stress responses [40]. These results highlight that exposure to stressful life events during pregnancy may be a better predictor of infant weight gain than postpartum stress. Infants with alterations in their own physiologic stress response could have rapid weight gain through biological (impaired hypothalamic-pituitary-adrenal function) or behavioral (increased use of food to soothe due to infant temperament) mechanisms [41].

While perceived stress in mothers was not associated with the infants’ stress response, it is still worth exploring as a potential moderator of the relationship between exposure to stressors and infant weight gain. The MAMAS study reported an interaction with high perceived stress increasing the effect of stressful life events on infant outcomes. Similarly, another study conducted among predominantly Black women found that perceived stress increased the risk of preterm birth only for women who experienced high social disorder [42]. Together, these studies suggest that perceived stress may be important in the context of other measures of stress (such as the number of stressful life events) to predict infant outcomes. While this secondary data analysis was limited to postpartum perceived stress, future studies looking to explore the impact of maternal stress on infant weight gain should consider maternal assessment during the prenatal period and collecting multiple measures of stress.

Perceived stress was slightly lower in this sample with limited variability compared to other studies with similar populations (mean 12.8 compared to other studies with means ~16 out of 40) [29,43,44,45]. The lower score seen in our population could be explained by several reasons. First, it could reflect true lower levels of perceived stress. Second, given that PSS is self-reported, participants could be under-reporting how stressed they truly feel due to social-desirability bias. For example, one study conducted among 355 Black women in the South found similar low PSS scores, which were attributed to the cultural norms described in the Superwoman Schema, that Black women experience an obligation to manifest an image of strength and minimize reports of stress [44,46,47]. More research is needed to elucidate stress perception in this population and to determine if other measures of stress show alternative associations. 

The prevalence of rapid weight gain in infancy seen in this sample (47%) is similar to what other studies conducted with racially and ethnically diverse infants have found. For example, among infants in the Boston Birth Cohort, rapid weight gain in infancy at one year of age was 38.5% for full-term infants (39–42 weeks gestational age) and 60.1% for early-term infants (37–39 weeks gestational age) [8]. Similarly, Felder et al. observed that the prevalence of rapid weight gain from birth to six months was 28% in a racially and ethnically diverse sample [13]. The prevalence observed in this study is markedly higher than the prevalence reported in predominately White, higher-income samples (13–19%) [10,12]. This highlights the importance of understanding the etiology of rapid weight gain in infancy, particularly in low-income and Black communities.

Although not the primary aim of this study, it is worth noting that the only variables significantly associated with rapid weight gain were birthweight-for-gestational-age z-score, gestational age at birth, and breastfeeding. Our results suggest that the risk for rapid weight gain in infancy is likely influenced by factors prior to birth (pre-conception/gestational risk factors associated with increased birth weight) and breastfeeding [11]. This contrasts with studies that have found that low birth weight was predictive of rapid weight gain [10,12]. It is possible that these results are conflicting because these studies looked at the rate of weight gain from birth which may capture catch-up growth for low-birth-weight infants, although rapid weight gain from birth to four-months has been shown to track through the first year of life for 95% of infants [8]. Consistent with the previous literature, length of time breastfeeding was associated with decreased risk of rapid weight gain, with breastfeeding for the recommended 12 months being associated with a lower risk of rapid weight gain [10,48]. Other predictors of rapid infant weight gain reported in the literature include infant sex, firstborn status, Black/other race, Medicaid status, maternal cigarette use, formula use, and feeding on a schedule [10,12,48]. This study did not find an association between sex, race, or income/employment status and rapid weight gain. More research is needed to better understand what predicts and causes rapid infant weight gain so that prevention efforts can be targeted.

The limitations of this study are important to note. One limitation is the high proportion of missing data, particularly on the exposure variable. Although we attempted to account for this missing data using multiple imputations with variables associated with perceived stress and missingness, it is possible that an unobserved variable influenced stress perception and, therefore, was not accounted for in the imputation model. Another limitation is that the sample was limited to two prenatal clinics in one state in the southeastern US. Therefore, results may not generalize to other populations. However, this study had many strengths, including longitudinal data collection, allowing us to establish temporality between the exposure and outcome. Secondly, there were objective anthropometric data collected by trained staff at all timepoints.

## 5. Conclusions

While rapid weight gain in infancy was prevalent in this sample of predominately Black infants in the Southeastern US, perceived stress in mothers did not influence rapid infant weight gain. Although our hypothesis was not supported, it is still worth examining if maternal perceived stress influences infant weight gain in the context of other measures of stress or during the prenatal period. While exposure to stress is not easily modifiable, interventions exist to decrease maternal stress perception and may play an important role in childhood obesity prevention. Future studies with longer follow-up periods should explore when and how maternal stress influences the development of childhood obesity so that intervention efforts can be targeted.

## Figures and Tables

**Figure 1 ijerph-19-05743-f001:**
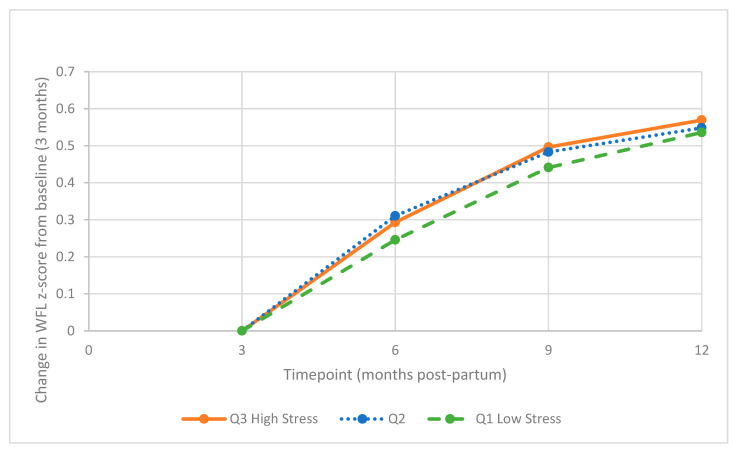
Mean Change in WFL Z-score by Baseline PSS Tertiles. Abbreviations: PSS, Perceived Stress Scale; WFL, weight-for-length, Q, quantile.

**Table 1 ijerph-19-05743-t001:** Characteristics of the Nurture Cohort by Baseline Perceived Stress Score (PSS)Tertiles *.

Variables N (%) or Mean ± SE	Full (*n* = 666)	Low Stress (*n* = 227)	Moderate Stress (*n* = 214)	High Stress (*n* = 224)	*p*-for-Trend
*Maternal*					
Age	27.3 ± 0.2	27.4 ± 0.5	27.4 ± 0.5	27.0 ± 0.4	0.55
Race					0.23
Black	477 (71.6)	168 (74.3)	144 (67.0)	165 (73.2)	
White	128 (19.3)	46 (20.4)	48 (22.2)	35 (15.4)	
Other **	59 (8.9)	12 (5.3)	23 (10.7)	25 (11.4)	
Ethnicity					0.40
(Hispanic/Latinx)	43 (6.5)	14 (6.0)	11 (5.3)	18 (8.2)	
Overweight or Obesity Pre-pregnancy	393 (59)	141 (62.3)	124 57.9)	128 (56.8)	0.43
BMI < 25 Pre-pregnancy	273 (41)	85 (37.7)	89 (41.5)	98 (43.8)	
Prenatal diet quality (AHEI)	42.2 ± 0.5	42.2 ± 0.9	42.3 ± 0.9	42.2 ± 0.9	0.96
Married or living with partner	380 (56.7)	133 (58.4)	124 (58.1)	123 (54.7)	0.51
Highschool degree or greater	531 (79.7)	185 (81.5)	175 (81.7)	171 (76.1)	0.31
Unemployed (looking for work)	125 (18.8)	37 (16.4)	47 (22.0)	41 (18.4)	0.27
Low Income(<20,000/year)	399 (59.9)	133 (58.5)	123 (57.2)	143 (63.9)	0.33
Number of people living in the home	3.5 ± 0.1	3.6 ± 0.1	3.4 ± 0.1	3.5 ± 0.1	0.61
Perceived Stress Score ***	12 ± 0.3	5.3 ± 0.2	12.3 ± 0.1	20.7 ± 0.4	<0.01
*Infant*					
Gestational age at birth	38.6 ± 0.2	38.7 ± 0.1	38.7 ± 0.1	38.4 ± 0.1	0.16
Birth weight for gestational age z-score	−0.3 ± 0.1	−0.3 ± 0.1	−0.3 ± 0.1	−0.3 ± 0.1	0.64
Small for gestational age (<10th%ile)	65 (9.8)	25 (10.8)	19 (9.1)	21 (9.5)	0.65
Large for gestational age (>90th%ile)	65 (9.8)	22 (9.9)	22 (10.2)	21 (9.3)	0.73
Sex Male	341 (51.2)	114 (49.9)	112 (52.1)	116 (51.7)	0.64
Weeks Breastfed	14.8 ± 1.5	14.8 ± 1.3	15.7 ± 1.4	13.6 ± 1.3	0.56
Breastfed 6 months or greater	150 (22.5)	53 (23.5)	54 (25.3)	42 (18.9)	0.28
Breastfed 12 months	90 (13.5)	32 (13.9)	31 (14.5)	27 (12.2)	0.65
Rapid weight gain at 12 months	314 (47.1)	107 (46.9)	99 (46.4)	108 (48.0)	0.57

Trends in baseline demographic characteristics across PSS tertiles using linear regression for continuous variables and Cochran-Mantel-Haenszel Test for categorical variables. * Results pooled over 50 imputations. ** Other includes participants that identified as Asian, Native Hawaiian/Pacific Islander, American Indian or Alaska Native, more than one race, or other. ******* PSS scored from 0–40 with a higher score indicating higher stress.

**Table 2 ijerph-19-05743-t002:** Association of Baseline PSS Tertiles and Change in WFL z-score from Baseline.

Variable	β Estimate	95%CI	*p* Value
Timepoint			
6 months	Ref		
9 months	0.20	0.08–0.31	0.001
12 months	0.29	0.18–0.44	<0.001
PSS Tertiles			
Q1 (Low)	Ref		
Q2	0.06	−0.13–0.26	0.531
Q3 (High)	0.07	−0.151–0.29	0.527
PSS × Time			
Q2 × 9 months	−0.02	−0.18–0.13	0.772
Q2 × 12 months	−0.05	−0.20–0.10	0.498
Q3 × 9 months	0.01	−0.15–0.17	0.923
Q3 × 12 months	−0.01	−0.18–0.15	0.868

Values were calculated with multivariable adjusted, repeated subjects, mixed-effects model, adjusted for birth weight, and gestational age at birth.

**Table 3 ijerph-19-05743-t003:** Association of Baseline PSS Tertiles and Risk of Rapid Weight Gain * between 3–12 months.

Maternal PSS	PSS Range	Risk Ratio Exp (β)	95%CI	*p* Value
Continuous	0–39	1.00	0.99–1.02	0.776
Tertiles				
Low	0–9	Ref	Ref	Ref
Moderate	10–15	1.01	0.80–1.26	0.945
High	16–39	1.06	0.81–1.38	0.657

Values were calculated with multivariable adjusted log-binomial model adjusted for birthweight and gestational age at birth. * Rapid weight gain is defined as a change in WFL z-score > 0.67 representing upward crossing of percentile bands on WHO growth chart.

## Data Availability

The datasets analyzed during the current study are not publicly available to protect the privacy of the research participants. Data are available from SEBN on reasonable request.

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
