# Peer review of "Maternal Stress and Excessive Weight Gain in Infancy"

_ijerph, 2022, doi:10.3390/ijerph19095743_

Round 1

Reviewer 1 Report

Dear Authors, This study will be of interest for all of us working in the area, basic scientists and health professionals alike.  I acknowledge that it is always a challenge using a database as the choice of when and what data is collected is already fixed.  Comments line-by-line:

Abstract, Line 12, "Therefore this study explores the impact of maternal perceived stress on the change in weight-for-length .....in infancy."  Please add the specific time points (3, 6, 9 and 12 months postpartum) that the PSS data was collected.  It would be helpful and relevant to state that the life-events data was collected and the exact period over which that data covered in relation to the gestation of the index child. 

Abstract, Line 19, Please consider amending the sentence to "About one-half of mothers were overweight or obese prior to pregnancy....."

Results, Lines 184 - 186 Please clarify the description of the ethnic groups "...non-Hispanic/LatinX (93.5%), Black (71.5%)..." This looks like 22% are non-Hispanic/Latinx and not Black, so what were they?  Even if the only possibility is to state "Other or unknown", this should appear on table 1

Please also clarify if 41% of the participants whose weight was not classified on table 1 were all not overweight or obese - or whether that 41% were a mixture of "unknown" and not overweight or obese.   

Discussion, lines 245 - 247 This study did find....life events a woman experienced in pregnancy was....weight gain from birth to 6 months." Perhaps more discussion about this, including the literature on epigenetic mechanisms that may underpin this effect might be helpful. 

It would also be relevant in the discussion (before line 255 "Taken together ....) and / or comment within the limitations section (lines 297 - 306) on the timepoints that the PSS data was available for.    The maternal perceived stress (PSS) data from specific time points during gestation would have been extremely helpful for this study.  The results associated with life events provide a strong argument for this being included in any future study design.     

Reviewer 2 Report

This study is interesting. The authors have pretty well described the results from the secondary data. The number of subjects was sufficient. However, the result didn't get support their hypothesis. Please find attached my comment in the PDF text.
